# Three-Dimensional Imaging of Circular Array Synthetic Aperture Sonar for Unmanned Surface Vehicle

**DOI:** 10.3390/s22103797

**Published:** 2022-05-17

**Authors:** Sai Zeng, Wei Fan, Xuanmin Du

**Affiliations:** 1National Key Laboratory of Science and Technology on Underwater Acoustic Antagonizing, Shanghai 201108, China; fanwei@csic726.com (W.F.); duxuanmin@csic726.com (X.D.); 2Shanghai Marine Electronic Equipment Research Institute, Shanghai 201108, China

**Keywords:** circular SAS, circular array SAS, sparse Bayesian learning, 3D imaging, USV

## Abstract

Synthetic aperture sonar (SAS) and interferometric synthetic aperture sonar (InSAS) have a range layover phenomenon during underwater observation, the AUV-mounted circular synthetic aperture sonar (CSAS) system, that insonifies targets using multiple circular scans that vary in height and can eliminate the layover phenomenon. However, this observation method is time-consuming and difficult to compensate. To solve this problem, the circular array synthetic aperture sonar (CASAS) based on the equivalent phase center was established for unmanned surface vehicles. Corresponding to the echo signal model of circular array synthetic aperture sonar, a novel three-dimensional imaging algorithm was derived. Firstly, the echo datacube was processed by signal calibration with near-field to far-field transformation and grid interpolation. Then, the sparse recover method was adopted to achieve the scattering coefficient in the height direction by sparse Bayesian learning. Thirdly, the Fourier slice theorem was adopted to obtain the 2D image of the ground plane. After the reconstruction of all height slice cells was accomplished, the final 3D image was obtained. Numerical simulations and experiments using the USV-mounted CASAS system were performed. The imaging results verify the effectiveness of the 3D imaging algorithm for the proposed model and validate the feasibility of CASAS applied in underwater target imaging and detection.

## 1. Introduction

Synthetic aperture sonar (SAS) is a well-established powerful underwater remote-sensing technique [1,2,3,4]. SAS systems could transmit acoustic pulses while moving along a trajectory and coherently combine the backscattered echoes to yield high-resolution images of the objects [5,6]. However, due to the platform of SAS moving in a straight trajectory, multiple acoustic scatterers may map to the same pixel in a beamformed SAS image. The SAS and synthetic aperture radar (SAR) community describe this phenomenon as range layover [4]. SAS can only obtain the projection of the real 3D scene on the 2D slant plane, but not the 3D image of the observed scene.

In order to obtain the 3D information of the observed scene, the researchers proposed the interferometric synthetic aperture sonar (InSAS) technology [7]. The InSAS imaging technology can obtain two 2D complex images under different viewing angles by using an interferometric array sonar system and then acquires the height information of the target by performing interference processing on the 2D complex images. However, the InSAS sonar imaging technology is not a real three-dimensional imaging technology, because it can only obtain the height of the target but cannot achieve the resolution of the target along the height direction. In addition, since InSAS processing assumes that each two-dimensional resolution cell contains only one scatterer in the height direction, when there are multiple scatterers in the height direction, the InSAS technology can only obtain the average height of scatterers. In the other words, InSAS also has a range layover phenomenon.

One approach for addressing range layover phenomenon is circular synthetic aperture sonar (CSAS) technology [4,8,9,10,11,12,13,14]. CSAS can obtain echo data in all directions of the target by a circular measurement trajectory. Based on the principle of computed tomography [15,16,17], the backscattered echoes are coherently processed to obtain the target image. CSAS imaging resolution is better than SAS and has three-dimensional imaging capability. Although CSAS has the ability of three-dimensional imaging, the lack of the wavenumber space spectrum in the height direction leads to poor height resolution of the three-dimensional image and serious conical side lobes. The method for improving the 3D imaging quality of CSAS is to erect a multidimensional synthetic array by conducting repeat passes at multiple altitudes or ranges and coherently combining the backscattered signals [4]. Such an approach has been performed and reported previously in the Synthetic Aperture Radar (SAR) community, where it is commonly referred to as SAR tomography [18,19,20,21,22]. When the airborne and spaceborne SAR tomography demonstration system moves in a circular trajectory, it is also called circular SAR (CSAR) tomography. Theoretical and laboratory-based studies have been conducted regarding the feasibility of this approach in the SAS community [11,23]. Recently, autonomous underwater vehicle-based (AUV) experiments have been performed to verify the validity of the method. The AUV-mounted sonar system insonifies targets using circular scans varying in height or radius, then coherently combines the backscattered signals to obtain 3D imaging of the target [4].

However, the synthesized multipass array will be irregular in the AUV-mounted sonar system due to navigation errors caused by currents [24,25,26]. This problem can fail to coherently combine the backscattered signals, which leads to unsuccessful attempts to obtain 3D imaging of the target. On the other hand, the AUV-mounted sonar system would surface and frequently obtain a new GPS lock for correcting underwater navigation trajectory during multipass measurement [25]; this is time-consuming and errors in the GPS localization would often cause a discontinuity in the final multipass aperture. This paper proposes a novel method to overcome these difficulties using the Unman Surface Vehicle (USV)-mounted array in height direction sonar system, which insonifies targets using a single circular scan, then coherently combines the backscattered signals to obtain the 3D imaging of target. Similar to the definition of Circular Array SAR (CASAR) in the literature [27], the USV-mounted array in height direction can be defined as Circular Array synthetic aperture sonar (CASAS). The structure of this paper is organized as follows: in Section 2, the CASAS system configuration, parameters, signal model, and data acquisition are introduced. In Section 3, the signal calibration, scattering coefficient reconstruction along the height direction, 3D imaging reconstruction procedure, and point spread function of CASAS are presented. The corresponding simulation and experimental results are shown in Section 4. Section 5 concludes the paper.

## 2. CASAS System

### 2.1. CASAS Configuration

In the SAS community, there are two kinds of transmitting-receiving models for sonar sensor array: single-transmitter multiple-receiver and multiple-transmitter multiple-receiver models. Each sensor transmits a signal and receives an echo independently based on the equivalent phase center principle. The imaging geometry of CASAS is shown in Figure 1, where the sonar system consists of a vertical line array operating at a *β* angle to the *Z*-axis. The sensor array moves along a circular trajectory with a radius of *R*_0_. Assuming that there are a total of *M* equivalent sensors whose *Z*-axis coordinates are $z_{m},\;m=1,\ldots ,M$, the distance of each equivalent sensor is *d*, the height of CASAS is *H*, and the circular trajectory of the first equivalent sensor is treated as the reference trajectory. When the CASAS sonar system moves to the azimuth angle *θ*, the instantaneous coordinate of the equivalent sensor *m* is ($R_{0}-md\cdot \sin\beta \cdot \cos\theta ,R_{0}-md\cdot \sin\beta \cdot \sin\theta ,z_{m}$), where $z_{m}=md\cdot \cos\beta +H$. It is assumed that the target scattering is isotropic and the sensors can illuminate the region of interest during the entire sampling time.

Generally, the target in the imaging scene can be approximated as a sum of several independent and non-directional scattering centers in SAS imaging. Consequently, the echo of the target can be expressed as the superposition of echoes of all scattering centers. Considering a general scattering center *P* with coordinate ($x_{P},y_{P,}z_{P}$), which is shown in Figure 1. The instantaneous distance from the scattering center to the equivalent sensor *m* can be given as
(1)$$R_{m}(\theta )=\sqrt{{\left[(R_{0}-md\cdot \sin\beta \cdot \cos\theta )-x_{P}\right]}^{2}+{\left[(R_{0}-md\cdot \sin\beta \cdot \sin\theta )-y_{P}\right]}^{2}+{\left(z_{m}-z_{P}\right)}^{2}}$$

Considering the target is in the far-field, the high-order terms can be neglected for the Taylor series expansion of the cosine term. The Equation (1) can be approximated as
(2)$${\tilde{R}}_{m}(\theta )\approx 2[\sqrt{R_{0}^{2}+z_{m}^{2}}-\frac{z_{m}z_{P}}{\sqrt{R_{0}^{2}+z_{m}^{2}}}-\frac{R_{0}}{\sqrt{R_{0}^{2}+z_{m}^{2}}}(x_{P}\cos\theta +y_{P}\sin\theta )]$$

### 2.2. CASAS Parameters and Signal Model

The pulsed linear frequency modulated (LFM) is appropriate for CASAS imaging. The transmitted signal can be presented as
(3)$$p(\tau )=rect(\frac{\tau }{T})\exp(j2\pi f_{0}\tau +j\pi \gamma \tau^{2})$$
where *τ* is the fast time, *T* is the pulse duration, *f*_0_ is the center frequency, and *γ* is the frequency modulated rate. For scattering center *P*, the echo received by equivalent sensors *m* can be expressed as
(4)$$s_{b}(\tau ,\theta ;m)=\sigma_{P}\cdot rect(\frac{\tau -R_{m}(\theta )/c}{T})\exp[j2\pi f_{0}(\tau -R_{m}(\theta )/c)+j\pi \gamma {\left(\tau -R_{m}(\theta )/c\right)}^{2}]$$
where $\sigma_{P}$ is the scattering coefficient of target *P* and *c* is sound speed.

For broadband signal, the matched filtering (MF) method will need a high sampling rate to satisfy the Nyquist sampling criterion [28], which will increase the data size remarkably. The bandpass sampling technique is able to overcome this difficulty [29].

### 2.3. CASAS Data Acquisition

The CASAS transmission signal *p*(*τ*) is the LFM signal and the echo signal is $s_{b}(\tau ,\theta ;m)$. The fast-time sampling frequency of the CASAS determines the number of range bins, and the pulse repetition frequency (i.e., the slow-time sampling frequency) determines the angle bins. The CASAS system received signals that are forming a time series 1D signal after the analog to digital converter (ADC). The fast- and slow-time samples of each equivalent sensor accumulated at each angle *θ*_1_, *θ*_2_, …, and *θ*_N_ are reshaped by the number of range bins. Hence, the sampling sequence of fast-time, slow-time, and equivalent sensor data are organized as a 3D complex time domain data matrix. As can be seen in Figure 2, the datacube is organized as $N_{\theta }\times N_{Range}\times N_{ch}$, where $N_{Range}$ is the number of fast-time sampling points, $N_{\theta }$ is the number of slow-time sampling data, and $N_{ch}$ is the number of receiver sensors.

## 3. CASAS Three-Dimensional Image Processing

### 3.1. Signal Calibration

In general, as in the two-dimensional case, the 3D datacube of circular array synthetic aperture sonar can be used to create 3D volumetric images simply by back-projection to voxels in the time–domain. However, this approach to beamforming is computationally burdensome and time-consuming. Two methods could be devised to reduce this burden. The first method beamforms data in the vertical dimension so that the Fourier slice theorem-based method can be applied to each horizontal slice [25]. Nevertheless, the Fourier slice theorem assumes that the data is acquired in the far-field of the object. For large objects or high-frequency synthetic aperture sonar systems, this can impose an experimentally unrealistic requirement on the radius of the synthetic aperture, forcing data to be acquired in the nearfield. Failure to meet the far-field criteria leads to distortions in the resulting image, angular spreading of narrow glints, and migration in an azimuthal angle of elastic features [4]. Referring to methods that have been suggested for SAR and ultrasonic imaging community, an algorithm can be proposed that allows 3D datacube measured in the near-field and then converted to the far-field, and such methods can be related to [29].
(5)$$\begin{array}{l} \Theta_{F-F}(\theta ,t)=F^{-(2)}[F^{\left(2\right)}{\left[\Theta_{N-F}(\theta ,t)\right]}_{n}\\ \cdot e^{-i[(4n^{2}-1)/2(4kR(\theta )+(4n^{2}-1)(4n^{2}-25)/6{\left(4kR(\theta )\right)}^{3}+(4n^{2}-1)(16n^{4}-114*4n^{2}+1073)/5{\left(4kR(\theta )\right)}^{5}]}] \end{array}$$
where $\Theta_{N-F}(\theta ,t)$ is the near-field backscattered acoustic data collect along a circular trajectory, $\Theta_{F-F}(\theta ,t)$ is the far-field converted data, *k* is the wavenumber, R(θ) is the radial diameter at θ, *n* is the azimuthal Fourier component, $H_{n}^{\left(1\right)}$ represents the *n*th-order Hankel function of the first type, and $F^{\left(2\right)}$ and $F^{-(2)}$ represent the forward and inverse two-dimensional Fourier transforms. This transformation makes the signals from near-field to far-field successful.

The second signal calibration of CASAS signal process is grid interpolation, which can be referred to [4]. The CASAS data is interpolated onto a common ground-plane grid in cylindrical coordinates, which means slant-range to ground-range interpolation. The grid has a uniform height of *h* = 0, spans a predefined *r* radius, the angularly spans 0 to 2π. The transformation from slant-range time units to ground-plane radial units can be shown in
(6)$$r=R(\theta )-\sqrt{{\left(tc/2\right)}^{2}-{\left(h_{CASAS}(\theta )\right)}^{2}}$$
where *R*(*θ*) and $h_{CASAS}(\theta )$ are the radius and height above the grid of the CASAS at *θ*. Following radial interpolation, the datacube is interpolated in the angular dimension to make the data regular in *θ*. The signal calibration can reduce the computational burden of CASAS 3D imaging [25].

### 3.2. Scattering Coefficient Reconstruction along the Height Direction

In order to increase the array size of CASAS, which is beneficial for the resolution in the height direction, the sparse hydrophone sensors array along the *Z*-axis is implemented. The density of the CASAS line array cannot satisfy the half-wavelength spacing condition, which cannot avoid the grating lobes. This makes the Fourier-based signal processing invalid [28]. Since there are limited dominated scattering centers in the height direction, the sparse reconstruction method can provide a solution to recover the scattering coefficient distribution [30,31]. It can extract the data corresponding to the same pixel cell of the multiple 2D complex images and rearrange them in a vector as the measurement data. For the pixel cell (*u*, *v*) of 3D imaging, the scattering distribution can be written as
(7)$$I(x_{u},y_{v},\varphi_{m})=\sum_{q=1}^{Q}\sigma_{q}\exp(j2Kz_{q}\sin\varphi_{m})$$
where $\varphi_{m}$ represents the pitch angle between the equivalent sensor and ground plane, *Q* is the number of grids divided along the height direction, the grids values are $z={\left[z_{1},z_{2},\ldots ,z_{Q}\right]}^{T}$, *T* denotes the transpose, *K* denotes signal wavenumber, and $\sigma_{q}$ represents the scattering coefficients of the target at grid *q*. Furthermore, the Equation (7) can be presented as the vector form $I_{u,v}(\varphi_{m})=G_{m}\sigma_{q}$, where $G_{m}=\exp(j2K\sin\varphi_{m}z^{T})$ and $\sigma_{q}=[\sigma_{1},\sigma_{2},\ldots ,\sigma_{Q}]$. For the *φ_m_*, $I_{u,v}(\varphi_{m})=G\sigma_{q}$, where $I_{u,v}(\varphi_{m})={\left[I_{u,v}(\varphi_{1}),I_{u,v}(\varphi_{2}),\ldots ,I_{u,v}(\varphi_{M})\right]}^{T}$ and $G={\left[G_{1},G_{2},\ldots ,G_{M}\right]}^{T}$. Considering that it will reconstruct the height scattering coefficient in the height direction pixel by pixel, the subscript *u*,*v* can be omitted in the following
(8)I=Gσ+N
where **N** is the additive white Gaussian noise, whose mean is zero and variance is *Ψ*. Since Q≫M, the solution of Equation (8) is underdetermined when assuming that the scattering center of the target behaves with spatial sparsity. Thus, assuming that *δ* is a proper bound, the Equation (8) can be transferred to an optimization problem as
(9)$$\underset{\sigma }{\min\left|\sigma \right|},s.t.\left\|I-G\sigma \right\|\leq \delta$$

In the SAS community, the number of dominated scattering centers are generally unknown, so it is impracticable to solve the Equation (9) based on the sparsity of signal. The sparse Bayesian learning (SBL) is an effective approach for sparse recovery. Compared with other *l*_1_-norm minimization algorithms, the SBL-based algorithm adopts iteration during the procedure to avoid convergence to the local minimum and produces a full posterior distribution as the solution [32,33,34,35]. Assuming that there are hyperparameter $\xi =[\xi_{1},\xi_{2},\ldots ,\xi_{Q}]$ governs σ, the prior probability of the distribution of scattering coefficient σ can be shown as
(10)$$f(\sigma |\xi )=\prod_{q=1}^{Q}\frac{1}{\sqrt{2\pi \xi_{q}}}\exp(-\frac{\sigma_{q}^{2}}{2\xi_{q}})$$

If the scattering coefficients are given, the probability of the measurement vector is determined by noise distribution, which can be presented as [30]
(11)$$f(I|\sigma ;\psi )={\left(\frac{1}{\sqrt{2\pi \xi_{q}}}\right)}^{M}\exp(-\frac{1}{2\psi }{\left\|I-G\sigma \right\|}^{2})$$

Considering the prior probability of σ and the conditional probability of I, the marginal probability of I can be written as
(12)$$\begin{array}{l} f(I|\xi ;\psi )=\int f(I|\sigma ;\psi )f(\sigma |\xi )d\sigma \\ =\frac{1}{\sqrt{\xi J+Gdiag(\xi )G}}{\left(\frac{1}{\sqrt{2\pi }}\right)}^{M}\exp(-\frac{1}{2}I^{T}{\left(\xi J+Gdiag(\xi )G\right)}^{-1}I) \end{array}$$
where J is the identity matrix. The posterior probability of σ can be written as follows based on Bayesian theorem
(13)$$\begin{array}{l} f(\sigma |I;\xi )=\frac{f(I|\sigma ;\psi )f(\sigma |\xi )}{f(I|\xi ;\psi )}={\left(\frac{1}{\sqrt{2\pi }}\right)}^{Q}\cdot \\ \frac{1}{\sqrt{\left|{\left[\psi^{-1}G^{T}G+{\left(diag(\xi )\right)}^{-1}\right]}^{-1}\right|}}\exp[-\frac{1}{2}\sigma^{T}(\psi^{-1}G^{T}G+{\left(diag(\xi )\right)}^{-1}-d\cdot J)\sigma ] \end{array}$$
where $d=\sigma^{-1}{\left[\psi^{-1}G^{T}G+{\left(diag(\xi )\right)}^{-1}\right]}^{-1}G^{T}I$. From Equation (12), it can be seen that the posterior probability is a Gaussian distribution with mean d. Furthermore, the variance is $\Xi_{\sigma }={\left[\psi^{-1}G^{T}G+{\left(diag(\xi )\right)}^{-1}\right]}^{-1}$. From Equation (13), the mean value can be treated as the estimated scattering coefficient, which is relative to the noise of variance *Ψ* and hyperparameter ξ. Both *Ψ* and ξ can be estimated by expectation–maximization (EM) steps in the traditional SBL process, nevertheless, the noise variance *Ψ* is a nuisance parameter that cannot frequently estimate with accuracy [32]. However, *Ψ* can be integrated out by introducting a gamma distribution based on reference [33], which can be written as
(14)$$f(\psi |c,d)=\frac{d^{c}\psi^{c-1}}{\Gamma (c)}\exp(-c\psi )$$
where *c* and *d* are small values and deterministic. $\Gamma (c)=\int_{0}^{\infty }x^{c-1}\exp(-x)dx$. The posterior distribution of σ is obtained in the form
(15)$$f(\sigma |I;\xi )=\int_{0}^{\infty }f(\sigma |I;\xi ,\psi )f(\psi |c,d)d\psi$$
where f(σ|I;ξ,ψ) is a multivariate complex Gaussian posterior distribution of σ, which can be shown as
(16)f(σ|I;ξ,ψ)=cN(σ|μ,Σ)
where $\mu =\psi \Sigma I^{H}I_{q}$, $\Sigma ={\left(\psi I^{H}I+diag(\xi )\right)}^{-1}$. Substituting Equations (13) and (15) into (14), the posterior of σ can be shown as
(17)$$f(\sigma |I;\xi )=\frac{\Gamma (c+Q){\left[1+{\left(\sigma -\bar{d}\right)}^{H}{\bar{\Phi_{\sigma }}}^{-1}(\sigma -\bar{d})/d\right]}^{-c+Q}}{\Gamma (c){\left(\pi d\right)}^{Q}\left|\bar{\Phi_{\sigma }}\right|}$$

The posterior distribution given by Equation (17) is a multivariate complex Student’s *t*-distribution [32], for which the mean and covariance are written as
(18)$$\begin{array}{l} \bar{d}=\bar{\Phi_{\sigma }}G^{T}I\\ \bar{\Phi_{\sigma }}={\left[G^{T}G+{\left(diag(\xi )\right)}^{-1}\right]}^{-1} \end{array}$$

From Equation (17), it can be seen that the estimate is merely a function of hyperparameter ξ rather than variance. Therefore, we have to estimate the hyperparameter ξ to obtain the scattering coefficient. Taking the EM approach of SBL into consideration, ξ can be obtained by maximizing the marginal probability of I [28]. The marginal likelihood function is obtained by integrating over the parameters σ and *Ψ*
(19)f(I|ξ)=∫f(I|σ;ψ)f(σ|ξ)f(ψ|c,d)dσdψ

Substituting Equations (10), (11) and (14) into (19) and evaluating this integral gives
(20)$$f(I|\xi )=\frac{d^{c}{\left(d+I^{H}\Omega^{-1}I\right)}^{-(M+c)}\Gamma (M+c)}{\Gamma (c)\pi^{M}\left|\Omega \right|}$$
where $\Omega =J+Gdiag(\xi )G^{H}$ and ${\left(\cdot \right)}^{H}$ denotes the conjugate transpose operator.

Taking the logarithm of f(I|ξ) and keeping only the terms that are dependent on hyperparameter ξ, we obtain the cost function
(21)$$F(\xi )=-(M+c)\ln(I^{H}\Omega^{-1}I)-\ln(\left|\Omega \right|)$$

Our goal is to maximize (21) with respect to the hyperparameter ξ. Differentiating F(ξ) with respect to $\xi_{q}$, then setting the result to zero yields
(22)$$\xi_{q}=\frac{1}{(M+c)\bar{d_{q}}/(d+I^{H}\Omega^{-1}I)+{\bar{\Phi }}_{\sigma_{q,q}}}$$
where $\bar{d_{q}}$ is the *q*th component of $\bar{d}$ and ${\bar{\Phi }}_{\sigma_{q,q}}$ is the *q*th diagonal of $\bar{\Phi_{\sigma }}$. Note that the hyperparameter ξ is a function of mean $\bar{d}$ and covariance $\bar{\Phi_{\sigma }}$. Meanwhile, Equation (18) indicates that mean $\bar{d}$ and covariance $\bar{\Phi_{\sigma }}$ are functions of ξ. Therefore, this optimization algorithm iterates between Equations (18)–(22) until convergence is achieved. Furthermore, it can obtain the scattering coefficient estimate by $\hat{\sigma }=\bar{d}$. As mentioned above, the processing procedure of SBL can be shown in Algorithm 1.
**Algorithm 1:** Procedure of SBL to Reconstruction Imaging along the height direction        **Input**:                 Matrix G, image vector in height direction;                     Initial hyperparameter ξ, parameter *c* and *d*;                     Convergence value ε;        **Output**:                  Mean $\bar{d}$; covariance $\bar{\Phi_{\sigma }}$;            1: **Begin**        2: Initialize input parameter, step *j* = 1;            3: **do**            4: computer ${\bar{d}}^{\left(j\right)}$ and ${\bar{\Phi_{\sigma }}}^{\left(j\right)}$ by Equation (18);            5: Update $\bar{d}$ and $\bar{\Phi_{\sigma }}$;            6: Calculate hyperparameter $\xi_{q}{}^{\left(j\right)}$ by Equation (22), *j* = *j* + 1;            7: **if**            8: $\left\|F{\left(\xi \right)}^{\left(j+1\right)}-F{\left(\xi \right)}^{\left(j\right)}\right\|<\varepsilon$            9: **end do**            10: **return**            11: $\bar{d}$,$\bar{\Phi_{\sigma }}$.        12: **End**

### 3.3. D Imaging Reconstruction Procedure

In order to reduce the computationally burdensome of 3D imaging and increase the resolution in the height direction, two methods are proposed to solve these problems. The first method beamforms data in the vertical dimension by SBL [35] followed by the application of the Fourier slice theorem-based method to each horizontal slice. Occasionally, the autofocus method can be used to improve the image quality of the horizontal slice 2D image [36,37,38,39], such as Shannon entropy metrics, generalized sharpness metrics, and contrast metrics. Finally, the 3D imaging of the data cube can be obtained by combining the 2D imaging along the height direction. The overall flowchart for 3D imaging is presented in Figure 3.

### 3.4. Point Spread Function of CASAS

The point spread function (PSF) defines the response of an imaging system to a point source [27]. For the CASAS imaging system, it is difficult to acquire the analyzed expression of CASAS PSF out of the image scene center. However, the PSF can be obtained by numerical calculation, which is shown in Figure 4. The main numerical calculation system parameters are listed in Table 1.

Figure 4a is the slice of CASAS 3D PSF in (0,0,0) and it can be seen that the target image focuses very well. Figure 4b is the slice of CASAS 3D PSF in (0,0,7), where the PSF is a circular lobe when it is out of the target. Figure 4c is the 3D PSF of CASAS which is shown in VAA3D software [40,41,42], and it can be seen that the PSF likes a “circular cone”. From Figure 4, it can be observed that the Peak Side Lobe Ratio (PSLR) is −7.96 dB and the Integral Side Lobe Ratio (ISLR) is −5.53 dB. Furthermore, the PSF has circular side lobes from the center.

## 4. Simulation and Experiment

### 4.1. Simulation Setup and Results

In this section, we present numerical simulations used to verify the effectiveness of the proposed 3D imaging model. The main simulation CASAS system parameters are listed in Table 2.

The simulated 3D target is shown in Figure 5. The 3D target consists of 116 scattering centers, for which the scatter coefficients are defined as 1. Figure 6 shows the simulated imaging scene, where the red cube is the 3D target, the black circular track is the trajectory of CASAS, the red pentagram defines the transmit sensor, and the blue points are receiver sensors.

The final 3D imaging simulation results using the 3D imaging reconstruction method are presented in Figure 7. Compared with Figure 5, it can be seen that the point scattering centers in the height z = −4 and the 3D target are reconstructed very well in the 3D space. Therefore, the simulation result demonstrates that the sparse recovery can reconstruct the height location of scattering centers, and the 3D imaging reconstruction method which is presented in this paper can reconstruct the 3D image of the target accurately.

### 4.2. Experiment Setup and Results

In December 2021, a series of CASAS experiments were conducted 20 m off the lakeside of Moganshan Lake, in Zhejiang Province, China. The utilized CASAS system has the operational frequency bands: the center frequency is 100 kHz and the band is 30 kHz. The length of the receiver array of CASAS is 1.2 m and the distance of each receiver is 2.5λ, where λ is the wavelength of center frequency. The main experimental CASAS system parameters are listed in Table 3.

The CASAS was mounted on the USV which was manufactured by Shanghai Marine Electronic Equipment Research Institute. The system can be shown in Figure 8a,b. The CASAS system can adjust the observation angle using the servo mechanism on the USV. The speed range of the USV is 0~3 kn. The Navigation devices on the USV include Real-Time Kinematic (RTK) Global Positioning System (GPS), Inertial Navigation System (INS), and Doppler Velocimeter (DVL). These navigation devices are beneficial for the platform motion calibration of CASAS. The transmit and receive array side of CASAS is perpendicular to the movement direction of USV during the trial. One of the main observation targets is shown in Figure 8c. It is a cube steel frame for which the size is 1.5 m × 1.5 m × 1.8 m, and there are several steel balls on each side of the steel cube which can be used to increase the target intensity.

The experimental results are shown in Figure 9, which contain the 2D slice imaging of the steel cube bottom, the reconstruction in the height direction at angle bin = 200, and the 3D imaging of steel cube. Comparing Figure 9a,b to Figure 8c, we can see that the 2D slice image has been reconstructed well, for the steel cube bottom, the steel cube side, and the ball can be seen clearly. Comparing Figure 9c to Figure 8c, we can see that the steel cube target has been reconstructed very well, and the 3D image is similar to a real steel cube. The bottom of the lake was also shown in Figure 9c. Therefore, the experimental results demonstrate the 3D imaging ability of the proposed imaging model and the sparse recovery algorithm.

## 5. Conclusions

Three-dimensional imaging is beneficial for verification and identification of underwater targets [43,44,45]. For the application of accurate 3D imaging for underwater targets, a novel 3D imaging process was proposed in this study. Due to the short wavelength of transmit waves, the sparse recovery method, which is based on sparse Bayesian learning, was applied to reconstruct scattering coefficients in the height direction with complex 2D images obtained by CSASA in each angle. After accomplishing the reconstructions of all angle bins, then, the high-resolution 2D imaging in horizontal slice was implemented by the Fourier slice theorem and the final 3D imaging was obtained by coherence stacking in the height direction. A numerical simulation was performed to verify the high-resolution imaging ability of the proposed imaging model and algorithm. Furthermore, the experiment had been conducted to verify the algorithm by a USV-mounted CASAS system in a lake. The experiment results verify the high-resolution imaging ability of the proposed imaging model and algorithm, which indicates that the USV-mounted CASAS system is suitable for 3D imaging of underwater targets.

## Figures and Tables

**Figure 1 sensors-22-03797-f001:**
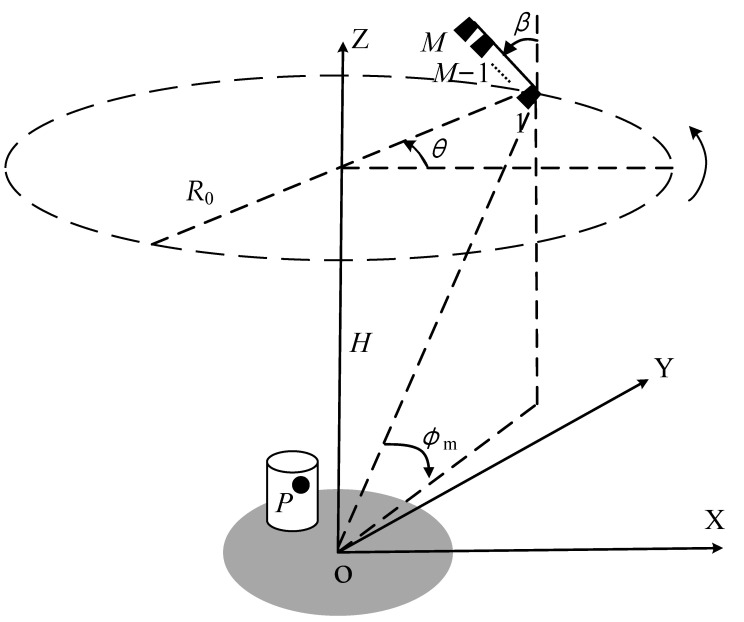
The imaging geometry of circular array synthetic aperture sonar.

**Figure 2 sensors-22-03797-f002:**
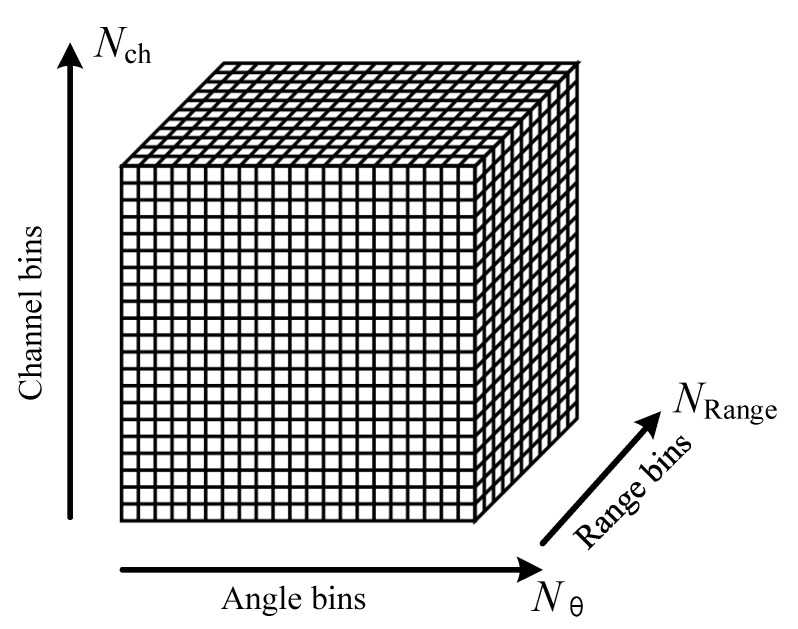
The datacube of circular array synthetic aperture sonar.

**Figure 3 sensors-22-03797-f003:**
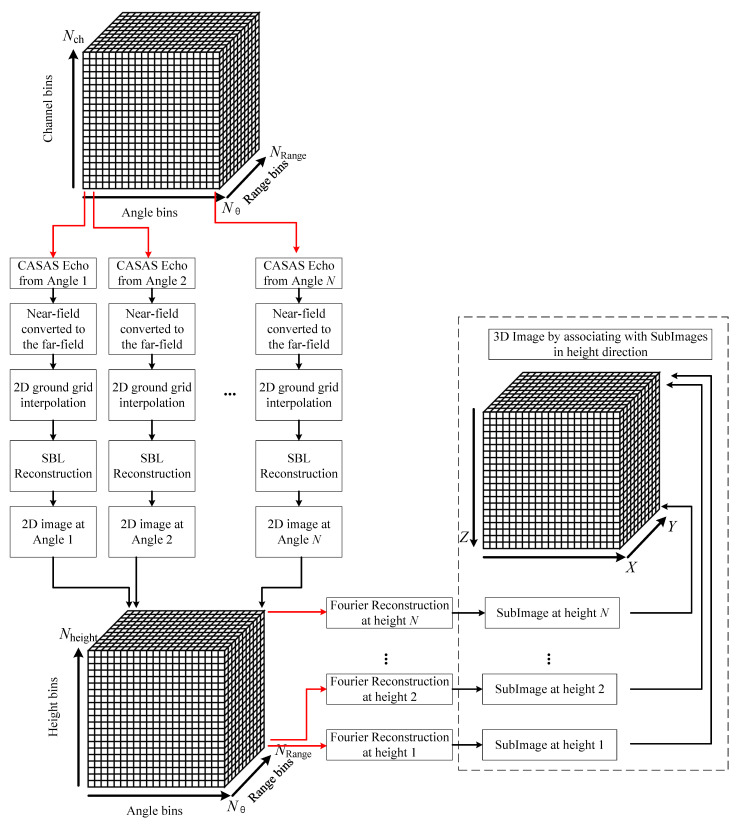
The 3D imaging flowchart of circular array synthetic aperture sonar.

**Figure 4 sensors-22-03797-f004:**
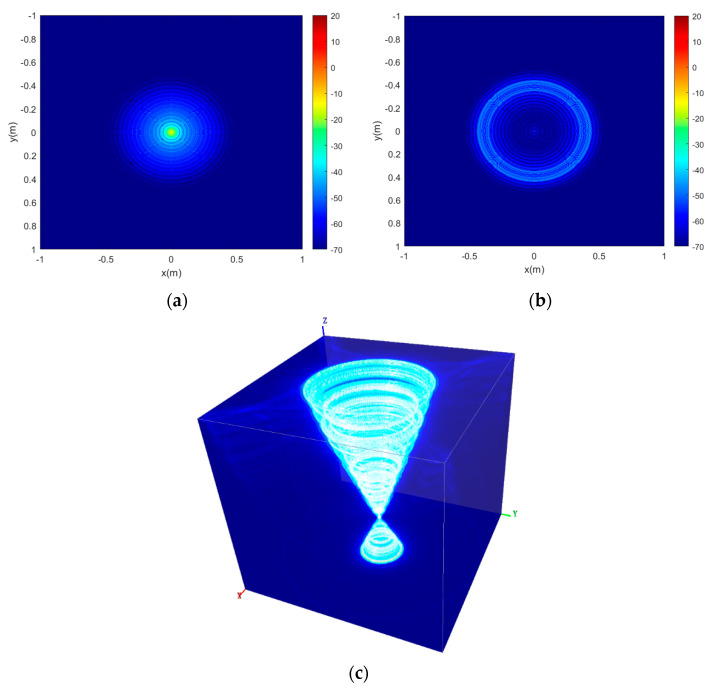
The 3D PSF of circular array synthetic aperture sonar. (**a**) Slice of CASAS 3D PSF in (0,0,0); (**b**) Slice of CASAS 3D PSF in (0,0,7); and (**c**) 3D PSF of CASAS.

**Figure 5 sensors-22-03797-f005:**
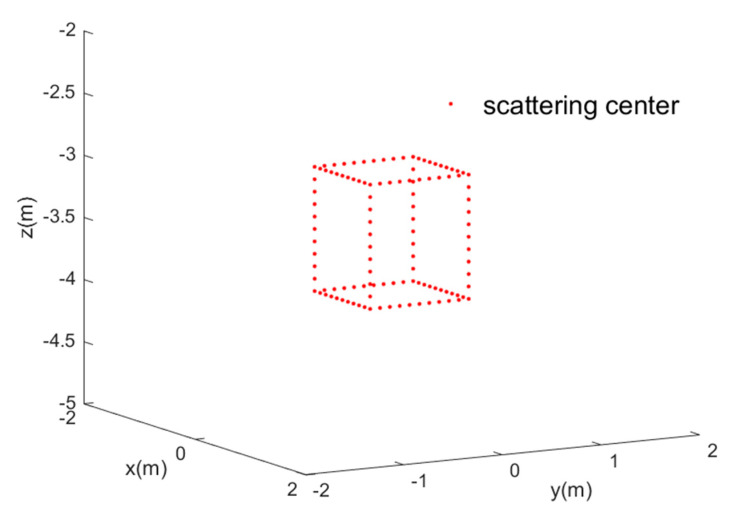
The simulated 3D target of circular array synthetic aperture sonar.

**Figure 6 sensors-22-03797-f006:**
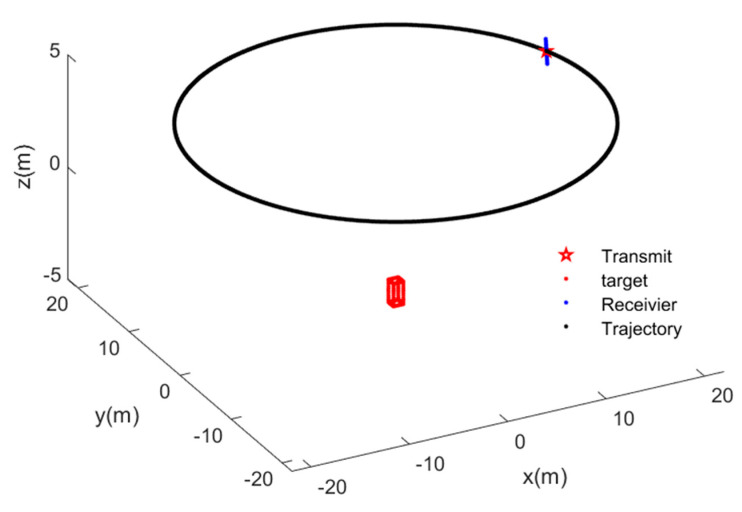
The image scene of circular array synthetic aperture sonar.

**Figure 7 sensors-22-03797-f007:**
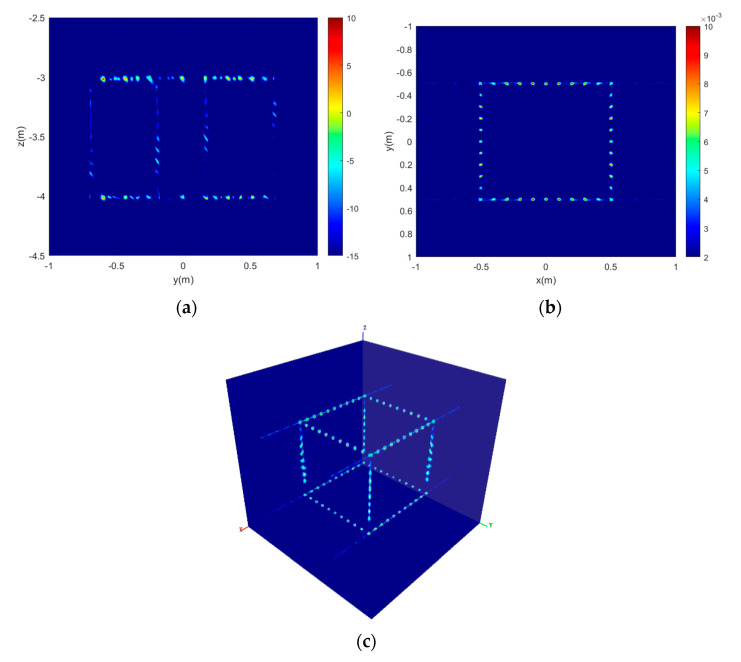
The 3D imaging simulation results of circular array synthetic aperture sonar. (**a**) the reconstruction in the height direction at angle 15°; (**b**) the 2D imaging result of horizontal slice at height z = −4; and (**c**) the 3D imaging result of the simulated target.

**Figure 8 sensors-22-03797-f008:**
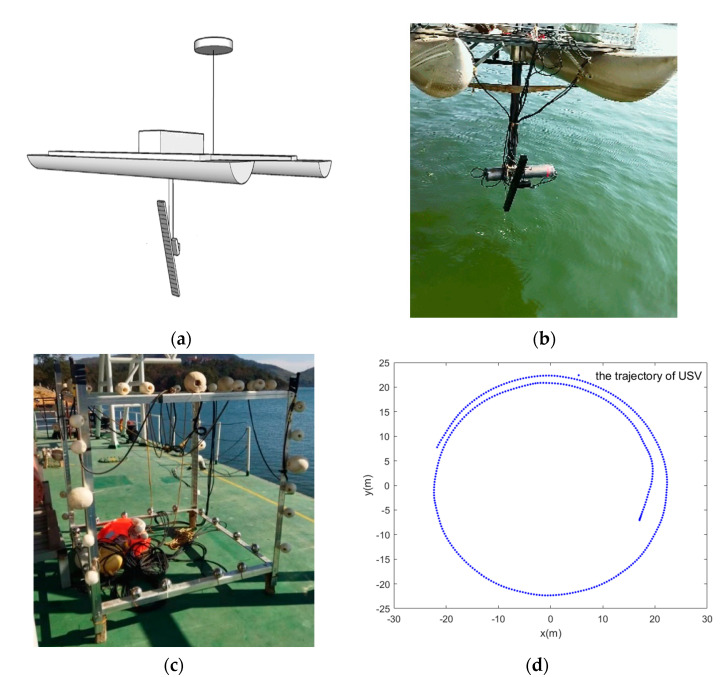
Experimental scenario and setup of circular array synthetic aperture sonar. (**a**) The schematic of CASAS installation in USV. It also shows the Navigation devices of USV, such as RTK GPS and INS. (**b**) The photo shows that the CASAS was installed in USV and the transmit and receive array side of CASAS perpendicular to the movement direction of USV during the trial. (**c**) The photo of the steel cube which was treated as experiment target. (**d**) The trajectory of USV in experiment which was recorded by GPS.

**Figure 9 sensors-22-03797-f009:**
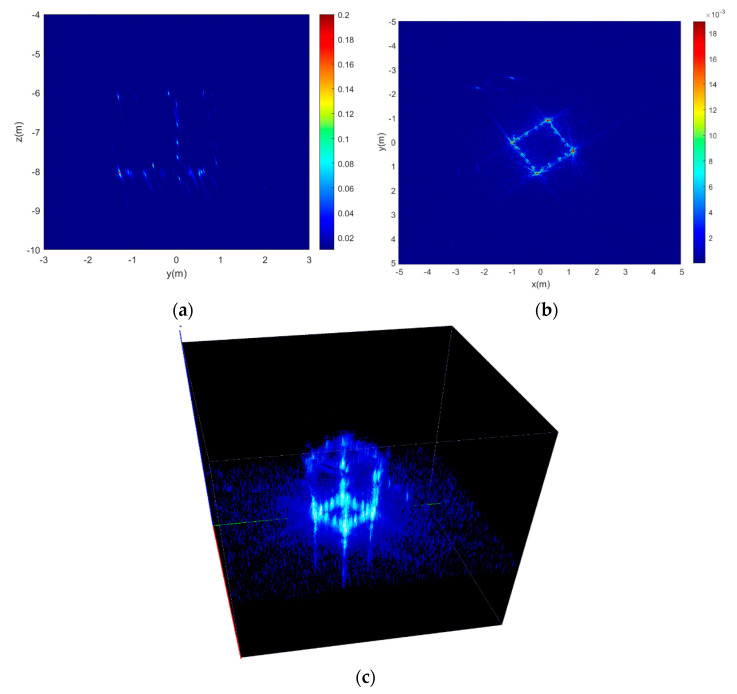
The 3D imaging experimental results of CSAS for steel cube target. (**a**) The reconstruction in the height direction at angle bin = 200; (**b**) the 2D slice imaging result at steel cube bottom; and (**c**) the 3D imaging result of the steel cube. It also contains the bottom image of lake.

**Table 1 sensors-22-03797-t001:** CASAS system parameters for PSF.

Parameters	Values
Point coordinate	(0,0,0)
Height of CASAS system	8 m
CASAS system Radius	20 m
Angle to *Z*-axis of CASAS	10°
Equivalent sensors	32
Center frequency	100 kHz
Band	30 kHz
Signal	LFM
Pulse duration time	40 ms
Pulse width	2 ms

**Table 2 sensors-22-03797-t002:** CASAS system parameters for 3D imaging simulation.

Parameters	Values
3D target	1 m × 1 m × 1 m (it consists of 116 scattering centers)
*Z*-axis coordinate of 3D target Bottom	−4 m
Height of CASAS system	4 m
CASAS system Radius	20 m
Angle to *Z*-axis of CASAS	10°
Equivalent sensors	32
Center frequency	100 kHz
Bandwidth	30 kHz
Signal	LFM
Pulse duration time	40 ms
Pulse width	2 ms
Angular sampling interval	0.2°

**Table 3 sensors-22-03797-t003:** CASAS system parameters for experiment.

Parameters	Values
Target	Steel cube 1.5 m × 1.5 m × 1.8 m
Target layout	At the bottom of lake
CASAS system Radius	~25 m
Angle to *Z*-axis of CASAS	10°
Equivalent sensors	32
Center frequency	100 kHz
Band	30 kHz
Array length of CASAS	1.2 m
Distance of receiver	2.5λ
Signal	LFM
Pulse duration time	100 ms
Pulse width	10 ms
Speed of USV	<1.5 kn
Navigation devices of USV	RTK GPS, INS, DVL

## Data Availability

Not applicable.

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
